# Single-Cell RNAseq Resolve the Potential Effects of *LanCL1* Gene in the Mouse Testis

**DOI:** 10.3390/cells11244135

**Published:** 2022-12-19

**Authors:** Jiangting Lu, Jinling Liao, Min Qin, Hui Li, Qingyuan Zhang, Yang Chen, Jiwen Cheng

**Affiliations:** 1Department of Urology, The First Affiliated Hospital of Guangxi Medical University, Nanning 530021, China; 2Center for Genomic and Personalized Medicine, Guangxi Key Laboratory for Genomic and Personalized Medicine, Guangxi Collaborative Innovation Center for Genomic and Personalized Medicine, Guangxi Medical University, Nanning 530021, China; 3Human Sperm Bank, The First Affiliated Hospital of Guangxi Medical University, Nanning 530021, China

**Keywords:** male infertility, sperm, single-cell RNAseq, LanCL1, transgenic mice

## Abstract

Infertility affects lots of couples, half of which are caused by male factors. The *LanCL1* gene is highly expressed in testis specifically, which might affect the development of sperms. In order to understand the potential functions of the *LanCL1* gene in the testis, this study was conducted with constructed transgenic LanCL1 knockout mice. The mouse breeding experiment, semen analysis and single-cell RNAseq of testicular tissue were performed. Results suggested that the *LanCL1* gene would significantly influence the reproduction ability and sperm motility of male mice. Single-cell RNAseq also confirmed the high expression of the *LanCL1* gene in the spermatocytes and spermatids. Downregulating the *LanCL1* gene expression could promote M2 macrophage polarity to maintain testicular homeostasis. Moreover, the *LanCL1* gene could affect both the germ cells and stromal cells through various pathways such as the P53 signaling and the PPAR signaling pathway to disturb the normal process of spermatogenesis. However, no effects of the *LanCL1* gene in testosterone synthesis and serum testosterone level were shown. Further studies are needed to discuss the mechanisms of the *LanCL1* gene in the various cells of the testis independently.

## 1. Introduction

Infertility might affect 15% of couples worldwide, which refers to the inability of naturally bearing children for 1 year or longer after marriage without contraception [1,2]. About 60% of infertility is caused by male factors [3]. From 1938 to 1990, the number of male sperms decreased by about 1% per year, from 113 × 10^6^ mL^−1^ in 1940 to 66 × 10^6^ mL^−1^ in 1990 [4]. Moreover, the rate of descent of sperms increased to 2.64% per year after 2000 [5]. Male infertility is not only an important health problem that hinders the continuation of the population, but is also closely related to malignant tumors. Infertile men with abnormal semen (such as reduced sperm motility, morphology, sperm count, etc.) were 20 times more likely to develop testicular tumor and 4–5 times more likely to develop an extra-gonadal tumor [6]. Therefore, clarifying the mechanisms of male infertility would be imperative.

It is suggested that the lanthionine synthetase components C (LanCL1) gene might be associated with male infertility. Early in 1998, Mayer et al. isolated LanCL1 from the human erythrocyte membrane [7,8]. Then, they emphasized that the *LanCL1* gene was highly expressed in the testis and brain, both with the blood-tissue barriers [9]. This hinted at the role of the *LanCL1* gene in the immune surveillance of these organs. In 2004, Nielsen et al. [10] discovered the high LanCL1 mRNA level in the meiotic division phase and early round spermatids and low-level in the elongating spermatids, which confirmed the vital function of the *LanCL1* gene in the germ cell differentiation. Additionally, The *LanCL1* gene was said to be important in the process of oxidative stress, which was also one of the mechanisms of male infertility [11,12]. In 2014, Huang et al. [11] identified the antioxidant activity of the *LanCL1* gene in the neuronal survival. In the prostate cancer (PCa), the *LanCL1* gene was also proved to protect the PCa cell from oxidative stress through suppression of the JNK pathway [13]. Meanwhile, we also found that oxidative damage in the brain of LanCL1 knockout mice could be alleviated by gut microbiota [14]. Recently, Huang et al. [15] performed a mechanistic study, which found direct antioxidant activity of the *LanCL1* gene in the mice spermatogenesis. However, no comprehensive landscape of all testicular cells and cell interactions after the *LanCL1* gene knockout on the single-cell level had been described. In order to understand the potential functions of the *LanCL1* gene in the testis, this study was conducted in the transgenic LanCL1−/− mice with the single-cell RNA-seq, mouse breeding experiment and semen quality assessment applied. Our study would elucidate the potential effects of the *LanCL1* gene in various testicular cells, which might pave the way for the further mechanistic investigation of the *LanCL1* gene in the male infertility on the single-cell level.

## 2. Methods and Materials

### 2.1. Animals

The LanCL1+/− mice on a C57BL/6J background was successfully constructed by the Shanghai Biomodel Organism Science & Technology Development Co., Ltd., Shanghai, China, with the CRISPR/Cas9 technology. All the mice were fed in the specific pathogen-free (SPF) conditions with constant ambient temperature and humidity and a 12 h light/dark cycle. Homozygous *LanCL1* gene knockout mice (LanCL1−/−) were acquired by copulation of heterozygous LanCL1 mice (LanCL1+/−). The identification method of mice genotypes was shown in our previous study [14]. Briefly, mouse tail tips were collected to acquire the genomic DNA with the routine phenol-chloroform method. Then, the DNA was used for the PCR amplification (94 °C for 5 min, 35 cycles of 94 °C for 30 s, 62 °C for 30 s and 72 °C for 1.5 min, then 72 °C for 5 min and 12 °C for holding) followed by agarose gel examination. Three PCR primers (P1: 5-GGAAATGCTTTAGGCAGACAG-3, P2: 5-GCAACTCCACCTGCTGACA-3, and P3: 5-CAGCGATGCCTGGAATGT-3) were combined to define the genotype of mice. Wild type mice were characterized by the P1 + P2 band of 1371 bp and the P1 + P3 band of 690 bp. And homozygous knockout mice (LanCL1−/−) were identified by only a P1 + P2 band of 500 bp and no P1 + P3 band (Appendix A). The animal experiments were approved by The Medical Ethics Committee of The First Affiliated Hospital of Guangxi Medical University.

### 2.2. Mouse Breeding Experiment

In order to evaluate the function of the *LanCL1* gene in the reproductive ability, the mouse breeding experiment was performed. Beginning at 6 weeks, LanCL1−/− and wide type (LanCL1+/+) male mice were intercrossed with the wide-type female mice, one by one. Once the vaginal plug was identified, the new LanCL1+/+ female mice were replaced in the cage after 2 days of rest. Four female mice would be used in every mating cage. Six repeated group experiments were carried out The pregnancy rate of female mice, occurrence time of vaginal plug, litter size and the gender of neonatal mice were recoded.

### 2.3. Semen Quality Analysis

The cauda epididymides were collected from the LanCL1−/− and LanCL1+/+ adult mice (8 weeks) and placed in the pre-warmed D-PBS, then transferred to a 1.5 mL tube with pre-warmed M16 media (1.71 mM CaCl_2_. 2H_2_O, 1.37 mM MgSO_4_, 4.78 mM KCl, 1.2 mM KH_2_PO_4_, 25 mM NaHCO_3_, 94.66 mM NaCl, 0.4% BSA, 5.5 mM D-glucose, 28.2 μM Phenol Red. Na, 0.33 mM Pyruvic Acid. Na, 26.1 mM DL-Lactic Acid. Na). The cauda epididymides were dissected to release the sperms into the buffer adequately. The sperm motility and concentration were assessed with the computer-assisted sperm motility analysis system (Integrated Semen Analysis System; Hview, Fuzhou, China). A total of 5 μL of semen was placed in a chamber of a cell slide. The slide was visualized under the 20× objective of the microscope. During the analysis, 10 fields per sample were evaluated, with >200 individual trajectories recorded. However, this record was not always possible for the LanCL1−/− males due to low sperm counts.

### 2.4. Quantitative Real-Time Polymerase Chain Reaction (qPCR)

Testis’s tissue was cut from the 8-week LanCL1−/−, LanCL1+/+ and LanCL1+/− mice. On the basis of manufacturer’s protocol, total mRNA was isolated from testis using RNeasy Mini Kit (Qiagen, Dusseldorf, Germany). RNA was reverse transcribed to cDNA using the random primer kit (Invitrogen, Carlsbad, CA, USA, NO. 48190011). QPCR was performed on Roche LightCycler 96 System (Roche, Basle, Switzerland), using SYBR Green Master Mix (Roche) with three replicates. The Gapdh (forward: 5′-TGTGAACGGATTTGGCCGTA-3′; reverse: 5′-GGCCTCACCCCATTTGATGT-3′) and the *LanCL1* gene primers (forward: 5′-TTGCCTTGCTTTTCGCTGAC-3′; reverse: 5′-GGCAGTCACATCCCATTCCA-3′) were designed.

### 2.5. Western Blotting

The protein was also extracted from the testis’s tissue of adult mice with RIPA buffer. The concentration of protein was quantified with the bicinchoninic acid (BCA) protein quantitative assay (Thermo, Waltham, MA, USA). Prepared protein was separated by 10% SDS-PAGE and transferred into PVDF membranes (Merck Millipore, Burlington, MA, USA). Then, membranes were blocked in 3% bovine serum albumin (BSA) with PBS. The anti-β-actin antibody (Cell Signaling Technology, Boston, MA, USA) and anti-LanCL1 antibody (Invitrogen, Carlsbad, CA, USA) were used to detect the protein expression according to the manufacturer’s protocol.

### 2.6. Immunofluorescence

The adult mouse testis was processed into tissue slices for immunofluorescence. Rabbit anti-LanCL1 (1:100) and Alexa Fluor 555-conjugated donkey anti-rabbit (1:200) were used to display the location of LanCL1 in the testis. Cell nuclei were stained with DAPI (1:1000); all the images were collected with the OLYMPUS microscope (BX53, OLYMPUS, Tokyo, Japan).

### 2.7. Testosterone Level

The blood was collected from three paired LanCL1−/− and LanCL1+/− mice, standing at room temperature for 1 h. Then, serum was extracted after centrifugation with 1000 rpm for 10 min. Serum testosterone levels were detected with the mouse enzyme-linked immunosorbent assay (ELISA) kit (R&D Systems, Minneapolis, MN, USA) according to the instructions. Briefly, working standards were prepared at room temperature. A total of 50 μL of Primary Antibody Solution was added to each sample well excluding the non-specific binding (NSB) wells and incubated for 1 h at room temperature. Each well was washed for four times and the plate was inverted a blotted. A total of 100 μL of standard, control, or sample was added to the plate wells and 50 μL of the Testosterone Conjugate was added to each well. The plate was incubated for 3 h at room temperature on a horizontal orbital microplate shaker. Then, each well was washed four times and the plate was inverted and blotted. A total of 200 μL of substrate solution was added to each well and incubate for 30 min in the dark. A total of 50 μL of stop solution was added to each well and thorough mixing was ensured. Finally, using a microplate reader (bio-tek, Winooski, VT, USA) set to 450 nm and 540 nm, the absorbance was measured. The concentration of testosterone was calculated based on the corresponding mean absorbance from the standard curve.

### 2.8. Single Cell RNA Sequencing for the Testis Tissue

The fresh testis was collected and washed with Dulbecco’s PBS (D-PBS) from two pairs of LanCL1−/− and LanCL1+/+ mice. And tissue digestion was processed in 15 min. The digestion of testis took place in three steps. Firstly, the tunica-free testis was incubated with an enzymatic digestion buffer I (1 mg/mL collagenase type IV, 0.3 mg/mL DNase I in HBSS) at 37 °C for 7 min with gentle oscillation. The suspensions were allowed to settle naturally and were then moved into tubes. Secondly, the tubules were re-suspended in an enzymatic digestion buffer II (1 mg/mL collagenase type IV, 1 mg/mL hyaluronidase, 0.4 mg/mL DNase I in HBSS) at 37 °C for 15 min. The suspensions were allowed to settle naturally, and were then moved into tubes. Lastly, the tubules were cut into small pieces and intubated with TrypLETMExpress (gibco, Waltham, MA, USA) at 37 °C for 5 min with gentle pipetting. Three supernatant tubules were filtered through 100 um filter. After centrifugation and removal of the supernatant, D-PBS was added; the cells were filtered through 40 um filter. The cell pellets were resuspended in D-PBS and cell numbers in each tube were counted, and the cells digested in the three steps were then mixed in proportion. The numbers of mix cells were counted using trypan blue staining.

Prepared single-cell suspension was loaded in a 10× chromium single-cell instrument (10× Genomics) based on the manufacturer’s protocol. Barcodes were used to mark every individual cell. PCR duplicates were recognized by the various unique molecular identifiers (UMIs). A 10× chromium single-cell 3′ library kit was used to construct the cDNA library. Finally, P5, P7, a sample index and read 2 (R2) primer sequence were added according to the manufacturer’s instructions.

## 3. Data Processing

The sequenced data was processed with Cell Ranger 6.1.1 “http://10xgenomics.com” (3 January 2022). After the continuous procedures of “cellranger mkfastq” and “cellranger count”, the gene expression matrices were generated. Then, the Seurat R package (Version 4.1.1) was used for the further quality control and dimensionality reduction analysis. Every gene was restricted to express in more than 10 cells. The number of genes in every cell was defined as >500 and <6000 to avoid low-quality cells and possible cell doublets. The percentage of mitochondrial genes was restricted to <10%. The integration of the four single-cell data followed the process of “SelectIntegrationFeatures”, “FindIntegrationAnchors” and “IntegrateData”. The expression matrix was log transformed and normalized. The percentage of mitochondrial genes was regressed out. A total of 30 principal components (PCs) were selected in the unified manifold approximation and projection (UMAP) dimensionality reduction analysis. Then, the function of “FindCluster” was performed with the resolution parameter set to 0.1. The differentially expressed genes (DEGs) were calculated with “FindAllMarkers” and “FindMarkers”.

### 3.1. GO Annotations

The “clusterProfiler” R package was used to perform the GO annotations. The inputted genes were converted into gene numbers with the “org.Mm.eg.db”. Then, the top ten “biological process” annotations were presented.

### 3.2. Gene Set Enrichment Analysis (GSEA)

Genes with Log2FC > 0.25 were defined as the DEGs, which were used in the GSEA based on the ‘clusterProfiler’ R package. The reference molecular signatures of mice (Mm.c2.cp.kegg.v7.1.entrez.rds) were downloaded from the online database https://bioinf.wehi.edu.au/software/MSigDB/ (15 January 2022). *p* < 0.05 was treated as the significantly enriched KEGG pathway.

### 3.3. Cell Communication

Cellchat “http://www.cellchat.org/” (15 February 2022) was used to discover the cell interaction before and after repressing the *LanCL1* gene expression. On the basis of the processes of cellchat, the function of “identifyOverExpressedGenes”, “identifyOverExpressedInteractions”, “projectData”, “computeCommunProb”, “computeCommunProbPathway” and “aggregateNet” with default parameters were performed to calculate the cell–cell communication networks and ligand-receptors communication.

## 4. Results

### 4.1. *LanCL1* gene Associated with the Male Reproductive Ability

On the basis of the BioGPS “http://biogps.org/#goto=welcome” (23 November 2021), we extracted the human and mice data of *LanCL1* gene expression in various organs. In the probe of 142701_a_at and 20219_a_at, the *LanCL1* gene was highest expressed both in human and mice, comparing to other tissues, such as lung, liver, gastrointestinal tract, etc., which suggested that the special function of the *LanCL1* gene in the testis. Therefore, we performed a mouse breeding experiment to further discuss the capacity of the *LanCL1* gene in the reproductive ability. Though without significant statistical difference (*p* = 0.084), the appearance of the vaginal plug seemed to be earlier in the wild type mice (2.33 ± 0.16 days, CV = 32.63%), comparing to the LanCL1−/− male mice group (3.00 ± 0.28 days, CV = 55.96%). Moreover, the litter size was significantly larger in the wild type mice (LanCL1+/+: 7.73 ± 0.41, CV = 25.00%; LanCL1−/−: 6.59 ± 0.28, CV = 19.68%; total CV = 24.09%; *p* = 0.027) (Table 1). However, the *LanCL1* gene could not influence the genders of neonatal mice.

### 4.2. Poor Semen Quality Present in the LanCL1−/− Mice

The testis and epididymis were collected from the LanCL1−/− and LanCL1+/+ mice to discuss the potential function of the *LanCL1* gene in the semen quality. Firstly, the absent *LanCL1* gene expression was confirmed on the RNA (LanCL1+/+: 3.18 ± 0.41, CV = 38.94%; LanCL1+/−: 1.12 ± 0.01, CV = 3.13%; LanCL1−/−: 1.00 ± 0.04, CV = 11.04%; *p* < 0.001) and protein levels (LanCL1+/+: 1.20 ± 0.00, CV = 0.33%; LanCL1+/−: 0.05 ± 0.03, CV = 85.8%, *p* = 0.001) in the testis again. (Figure 1a) Further, we also detected the sperm parameters and morphology. Comparing to LanCL1+/+ mice, the sperm concentration (LanCL1+/+: 48.45 ± 3.47, CV = 14.31%; LanCL1−/−: 37.18 ± 1.89, CV = 10.16%; total CV = 18.54%; *p* = 0.029) and motility (LanCL1+/+: 10.62 ± 2.89, CV = 54.45%; LanCL1−/−: 2.64 ± 0.78, CV = 59.19%; total CV = 87.42%; *p* = 0.037) were significantly lower in the LanCL1−/− mice (Figure 1b). However, no observable morphological abnormalities of the sperm presented after inhibiting the *LanCL1* gene expression (Figure 1c). These results suggested that the *LanCL1* gene would affect the male semen quality.

### 4.3. Single-Cell Analysis Discover the High Expression of Cir1 Gene in the LanCL1−/− Mice

In order to understand the function of the *LanCL1* gene in the testis, the single-cell RNAseq was conducted with four testes tissues from the LanCL1−/− and LanCL1+/+ mice. After rigorous quality control, 41,166 cells were left in the analysis. Then, 12 cell clusters were defined including spermatogonia (SPG), meiotic spermatocytes (SCytes), post-meiotic haploid round spermatids (STids), elongating spermatids, sertoli, myoid, macrophage (Mac) and leydig with specific cell markers [16] (Figure 2a,b). The *LanCL1* gene expression was highly expressed almost in all the germ cells comparing to stromal cells (Figure 2c). In these cells, Leydig cells had the lowest mRNA expression of the *LanCL1* gene. Oppositely, the higher mRNA expression of the *LanCL1* gene was shown in the STids, SCytes and elongating spermatids (Figure 2c). An immunofluorescence image confirmed that the *LanCL1* gene mainly expressed in the end stage of spermatogenesis (Figure 2d). In order to further investigate the function of *LanCL1* gene in the spermatogenesis, we calculated the DEGs from all the germ cell clusters between the LanCL1−/− and LanCL1+/+ mice. There were 47 and 32 up-regulated genes in the LanCL1−/− and LanCL1+/+ mice, respectively (Figure 2e). The GO annotation identified that the function of DEGs in the LanCL1−/− mice referred to fertilization. The up-regulated genes in the LanCL1+/+ mice were related to the spermatogenesis, such as “germ cell development”, “spermatid nucleus differentiation”, and “sperm DNA condensation” (Figure 2e). These annotations also suggested that the *LanCL1* gene was significantly associated with reproductive function. We also analyzed the DEGs between the LanCL1−/− and LanCL1+/+ mice in the germ cell with higher *LanCL1* gene expression (Figure 2f). In all the up-regulated genes in the LanCL1−/− mice, the Cir1 gene was identified. The Cir1 gene was said to suppress the activation of the Notch signaling pathway. This suggested that LanCL1 would maintain the homeostasis of the Notch signaling pathway to protect from the normal spermatogenesis.

### 4.4. *LanCL1* gene Associate with the Normal Sperm Motility in Germ Cells

As shown in the Venn diagram, the number of up-regulated genes in the cluster 7 cell was the most compared to other germ cells (Figure 2f). Moreover, the cell percent of cluster 7 was higher in the LanCL1+/+ mice, which suggested that this germ cell was greatly influenced by the *LanCL1* gene (Figure 3a). Therefore, the cluster 7 cell was extracted to be re-clustered into four cell groups. (Figure 3b) Among them, cluster 0 and 1 had the higher cell proportion from the LanCL1+/+ mice (Figure 3c). Compared to cluster 2 and 3, the DEGs of cluster 0 and 1 functioned as flagellated sperm motility and cilium movement. Oppositely, the function of DEGs in the cluster 2 and 3 are said to participate in the meiotic process (Figure 3d). These results hinted that the *LanCL1* gene would regulate the normal sperm motility and the development of cilium and flagellum.

### 4.5. *LanCL1* gene Influences the Cell Interactions Signaling within the Germ and Stromal Cells and Promotes the M2 Macrophage Polarity in Mouse Testis

We further discussed the effects of the *LanCL1* gene in the cell interaction signaling. Interestingly, the numbers of cell interactions decreased after *LanCL1* gene knockout (Figure 4a). Most of the intensity of the signaling pathway was similar in the LanCL1−/− and LanCL1+/+ mice. In the LanCL1+/+ mice, the PDGF, SEMA3, EPHB, EPHA, CD48 and SEMA6 signaling pathway were active solely. Oppositely, the GALECTIN signaling pathway was only active in the LanCL1−/− mice (Figure 4b). The ligands and receptors of the PDGF signaling pathway were located on the SCytes1 and Myoid Leydig cell, respectively (Figure 4c). The Pdgfa gene was only highly expressed in the SCytes1 and Scytes4 of LanCL1+/+ mice. The Pdgfra receptor was wildly expressed in the Myoid and Leydig cell, which suggested that the *LanCL1* gene would regulate Pdgfa gene expression. Additionally, the GALECTIN signaling was active after *LanCL1* gene knockout, in which the Lgals9 gene was up-regulated in the Leydig cell to influence the macrophage. Compared to LanCL1+/+ mice, the macrophage in the LanCL1−/− mice was presented to be M2 polarity, with many M2 macrophage maker genes expressed (Figure 4f).

### 4.6. *LanCL1* gene Regulated the P53 and PPAR Signaling Pathway Other Than the Gene Expressions of Testosterone Synthesis in the Leyding Cells

As shown in the cell interactions analysis, there were a number of interactions within the Leydig cell and other cell clusters. Moreover, Leydig cell might also promote the M2 macrophage polarity (Figure 4a,e). Therefore, understanding the features of the Leydig cell in the LanCL1−/− mice would also be important. In the differential expression analysis, there were much more DEGs in the stromal cells compared to germ cells, which also suggested the vital role of the *LanCL1* gene in the stromal cells although with lower *LanCL1* gene expression (Figure 5a). In the Leydig cell, 179 up-regulated and 275 down-regulated genes were shown after *LanCL1* gene knockout (Figure 5a). The results of GSEA prompted the activation of the PPAR signaling pathway (NES = 1.59, *p* = 0.048) and P53 signaling pathway (NES = −1.59, *p* = 0.045) in the LanCL1+/+ and LanCL1−/− mice, respectively (Figure 5b). However, the steroid biosynthesis signaling was not significant in the LanCL1+/+ mice, which suggested that the LanCL1 would not influence testosterone synthesis (Figure 5c). Most of the key rate-limiting enzymes such as Cyp17a1, Cyp19a1, Hsd3b1, Srd5a1, Srd5a2 and Hsd17b3, were also not remarkable different (Figure 5d). Serum testosterone level had no statistically difference between the LanCL1+/+ (0.97 ± 0.11, CV = 19.36%) and LanCL1−/− (2.60 ± 1.39, CV = 92.37%) mice (*p* = 0.361) (Figure 5e).

## 5. Discussion

Infertility is a worldwide disease and influences lots of families. Recently, male fertility and semen quality decreased dramatically. Various factors have led to this phenomenon. Analysis suggested that the *LanCL1* gene was highly expressed in the testis, which might have some effects in the male reproductive ability.

The *LanCL1* gene belongs to the wool thiocyanine synthase c-like protein family, encoding a 40 KDa protein. The polypeptide produced by LanCL1 has a certain antibacterial effect [17]. A previous study identified that there was a high expression of the *LanCL1* gene in testis and brain. These organs had the blood-tissue barriers. Therefore, they were speculated to have role in the immune surveillance [9]. Recent studies suggested that the *LanCL1* gene was an important glutathione binding protein expressed in the mammalian central nervous system, significant to neurodegenerative disease [18]. In 2012, Zhong et al. discovered that LanCL1 could inhibit cystathionine β-synthase, which is a critical enzyme for the reduced glutathione (GSH) synthesis and GSH-dependent defense against oxidative stress [19]. Many studies have also confirmed the antioxidant activity of the *LanCL1* gene in the neuronal system, PCa and gut [11,13,14]. Additionally, oxidative stress was also one of the mechanisms of male infertility [12]. Therefore, LanCL1 would be important in the reproductive system. However, no comprehensive landscape of all the testicular cell and cell interactions in the LanCL1−/− mice on the single-cell level had been described. In 2001, Mayer et al. [9] first described the expression of the *LanCL1* gene in the rat, which identified the strong signal of the *LanCL1* gene in germinal cells of the seminiferous tubules in the testis. Meanwhile, in the testis, the level of LanCL1 mRNA was differentiation-dependent and stage-specific. It was highly expressed in the early round spermatids and low in elongating spermatids [10]. These were consistent with our results of immunofluorescence and single-cell RNAseq. Recently, Huang et al. [15] performed a study about the LanCL1-regulated spermatogenic redox homeostasis, which had described the mechanisms of *LanCL1* gene in influencing the male subfertility. After *LanCL1* gene knockout, redox imbalance was observed in the testis with increased dihydroethidium-labeled ROS accumulation and a decreased NADPH/NADP ratio. In their study, they found the same results as ours about the decreased sperm motility and poor male fertility in the *LanCL1* gene knockout mice. The attenuated motility and fertility might be associated with the mitochondrial damage, oxidative stress and germ cell apoptosis, other than the acrosome reaction of spermatids for the absent *LanCL1* gene expression. Moreover, conditional increased *LanCL1* gene expression could attenuate the defects in spermatozoal concentration, motility and forward motility induced by oxidative damage. Additionally, we found that the Cir1 gene had up-regulated expression in *LanCL1* gene knockout mice. Cir1 is a co-repressor interacting with RBPJ, which mainly suppresses the activation of the Notch signaling pathway. Previous studies suggested that Notch signaling was important in the germ cell development in many other animals [20,21]. In Drosophila melanogaster, Notch signaling was identified to be essential for the survival of the germline stem cell lineage [22]. Reducing the Notch signaling would lead to the loss of germline cell [22]. Moreover, Notch signaling participated in the development of Sertoli cells and Leydig cells, which was important in the testis development [23,24]. In 2001, Hayashi et al. [25] suggested that Notch 1 and its ligand jagged 2 were important in the differentiation and maturation of germ cells both in the rat and human. Murta et al. [26] also identified that Notch signaling blockade could significantly increase the germ cell apoptosis and epididymis spermatozoa morphological defects. In the epididymis and spermatozoa, inhibiting Notch signaling would also decrease the sperm motility [27]. Therefore, the *LanCL1* gene would suppress the Cir1 gene expression to maintain the appropriate activity of Notch signaling, which further convoy the spermatogenesis.

On the basis of cell interactions analysis, we detected alterations in various signaling after *LanCL1* gene knockout. The PDGF and GALECTIN signaling were conspicuous. We found that the Pdgfa gene was higher expressed in the spermatocytes before *LanCL1* gene knockout. The animal experiment suggested that the Pdgfa gene knockout in mice would lead to the low level of circulating testosterone and spermatogenic arrest [28]. Additionally, in the breeding season, the Pdgfa gene was highly expressed in myoid and sperm cell, which also confirmed the important role of Pdgfa gene in the male fertility. Therefore, the *LanCL1* gene could regulate the expression of Pdgfa influencing the fertility. Additionally, we also identified the GALECTIN signaling was active in the LanCL1−/− mice. After *LanCL1* gene knockout, the Galectin 9 (Lgals9) gene expression was un-regulated; its receptor was located in the macrophages. Lgals9 coded a protein participating in modulating cell–cell and cell–matrix interactions. Lv et al. [29] suggested that the overexpression of Lgals9 could promote the M2-type macrophages polarity. In the LanCL1 knockout mice, ROS and inflammation increased [15]. In order to maintain body homeostasis, the anti-inflammatory macrophage would increase. Consistent with our results, the gene markers of M2-type macrophage were also highly expressed in LanCL1−/− mice.

Although with low *LanCL1* gene expression, many signaling pathways also emit from the Leydig cell. In order to understand the function of *LanCL1* gene in the Leydig cell, further analysis was performed. Although it is an important cell in testosterone synthesis, most of the key rate-limiting enzymes of testosterone synthesis such as Cyp17a1, Cyp19a1, Hsd3b1, Srd5a1, Srd5a2 and Hsd17b3 were not remarkably different in the LanCL1 knockout mice. Moreover, the serum testosterone level had also no statistically significant differences, which was also consistent with the results of Huang et al. [15]. Therefore, the *LanCL1* gene would not influence the function of testosterone synthesis in the Leydig cell. However, GSEA analysis suggested that the PPAR signaling pathway was inactive in the Leydig cell of LanCL1−/− mice, which might influence the normal spermatogenesis and sperm functions. As a member of the PPAR signaling pathway, peroxisome proliferator-activated receptor gamma (PPARγ) was identified to improve the motility, capacitation, acrosome reaction, survival and metabolism of sperm in humans and pigs [30,31]. Oppositely, the P53 signaling pathway was active after LanCL1 knockout. The P53 signaling pathway mainly participated in cell apoptosis. The P53 gene was said to maintain the normal morphology of sperm [32]. However, in the asthenozoospermic, the P53 gene expression was negatively correlated to sperm motility and concentration [33]. Moreover, the oxidative stress was also active in the asthenozoospermic [34]. Above all, we suggested that *LanCL1* gene would be important in the male spermatogenesis and reproduction.

## 6. Limitations

Our study would be the first to discuss the functions of the *LanCL1* gene in the various testicular cells and their interactions on the single-cell level, which might be beneficial to further research on the mechanisms of male infertility in single-cells. However, some limitations should also be emphasized as follows: (1) Some results of experiments such as semen analysis and testosterone level seemed to be variable in different mice. Therefore, repetitive experiments with many more mice should be considered in the further studies; (2) Results of detailed mechanisms were scarce in our study to verify the discovery of single-cell RNAseq.

## 7. Conclusions

The *LanCL1* gene was highly expressed in testis specifically, which might affect the normal function of testicular tissue. On the level of the single-cell, we confirmed the distribution of *LanCL1* gene expression mainly in the germ cells, which was associated with the reproduction ability and sperm motility of male mice. Meanwhile, *LanCL1* gene might promote the M2 macrophage polarity in mouse testis. Through various pathways, *LanCL1* gene also influenced both the germ cells and stromal cells, other than the synthesis and serum level of testosterone to affect the normal process of spermatogenesis.

## Figures and Tables

**Figure 1 cells-11-04135-f001:**
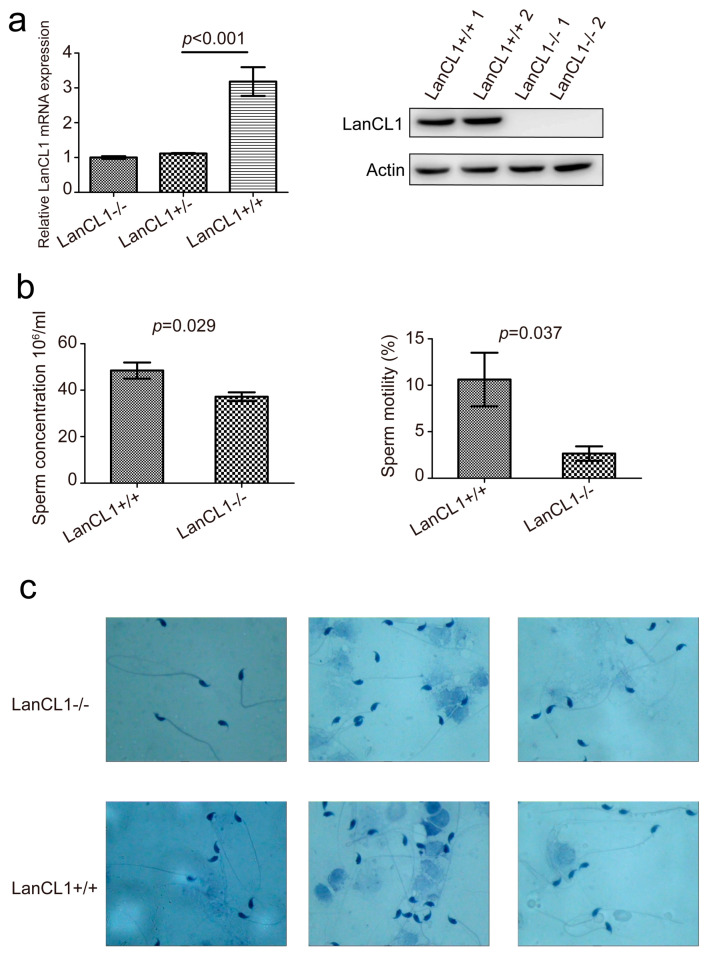
*LanCL1* gene knockout influences the semen quality. (**a**) The expression of mRNA and protein in the LanCL1+/+, LanCL1+/− and LanCL1−/− mice. (**b**) Parameters of semen quality in the LanCL1+/+ and LanCL1−/− mice. (**c**) Sperm morphology.

**Figure 2 cells-11-04135-f002:**
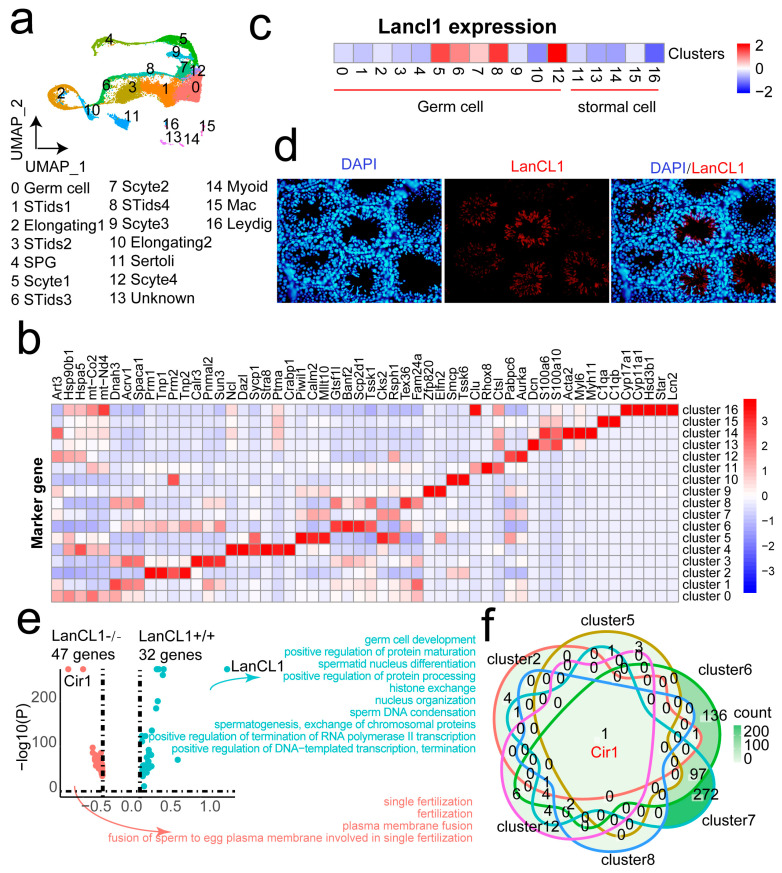
Single-cell RNAseq of the testis identified the high expression of the Cir1 gene in LanCL1−/− mice. (**a**) UAMP plot of the cell clusters. (**b**) Heatmap of the marker genes in the cell clusters. (**c**) The *LanCL1* gene expression in the cell clusters. (**d**) Immunofluorescence of the *LanCL1* gene in the testicular tissue. (**e**) The volcano plot of the DEGs in the LanCL1+/+ and LanCL1−/− mice. (**f**) Venn diagram of up-regulation in the Germ cells of the LanCL1−/− cell.

**Figure 3 cells-11-04135-f003:**
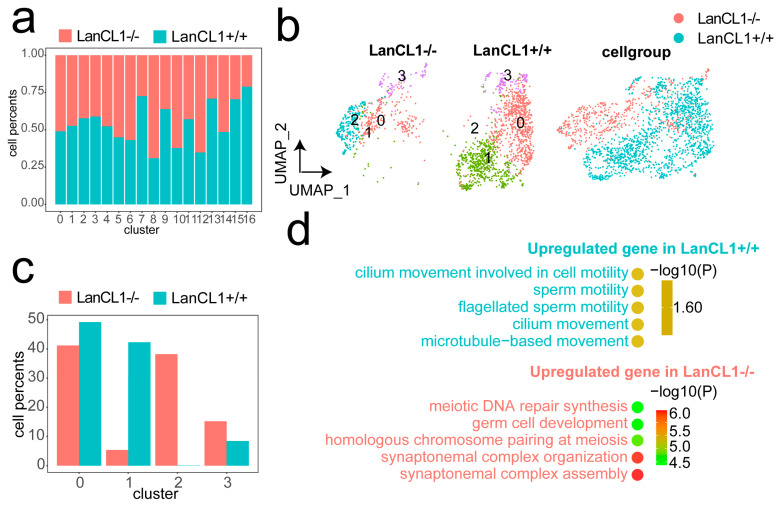
The *LanCL1* gene maintains the development of normal sperm motility in germ cells. (**a**) The percent of cell numbers in every cell cluster from LanCL1+/+ and LanCL1−/− mice. (**b**) UAMP plot of the cluster 7. (**c**) The cell percent of new clusters. (**d**) GO annotation of the DEGs in the LanCL1−/− and LanCL1+/+ mice.

**Figure 4 cells-11-04135-f004:**
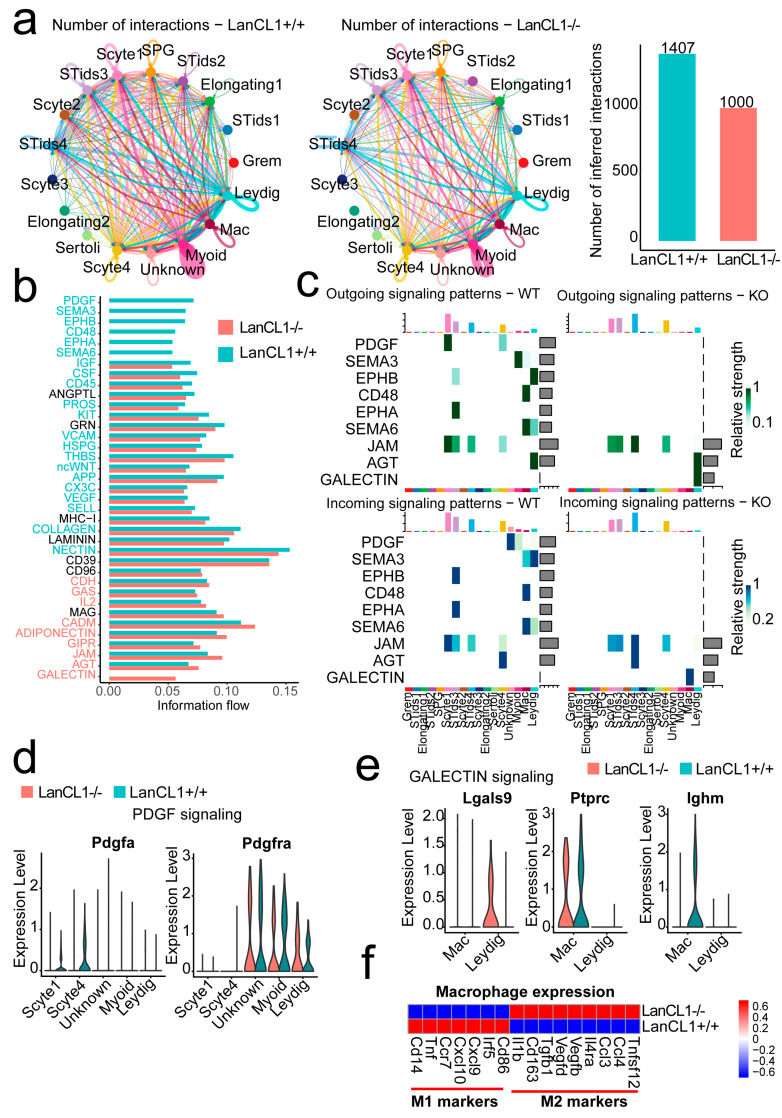
The cell interactions signaling in various cell clusters regulated by the *LanCL1* gene. (**a**) The cell communication before and after *LanCL1* gene knockout. (**b**) The alteration of signaling pathway. (**c**) Heatmap of the active signaling pathway in every cell clusters. (**d**) PDGF signaling pathway. (**e**) GALECTIN signaling pathway. (**f**) The M1 and M2 macrophage marker genes.

**Figure 5 cells-11-04135-f005:**
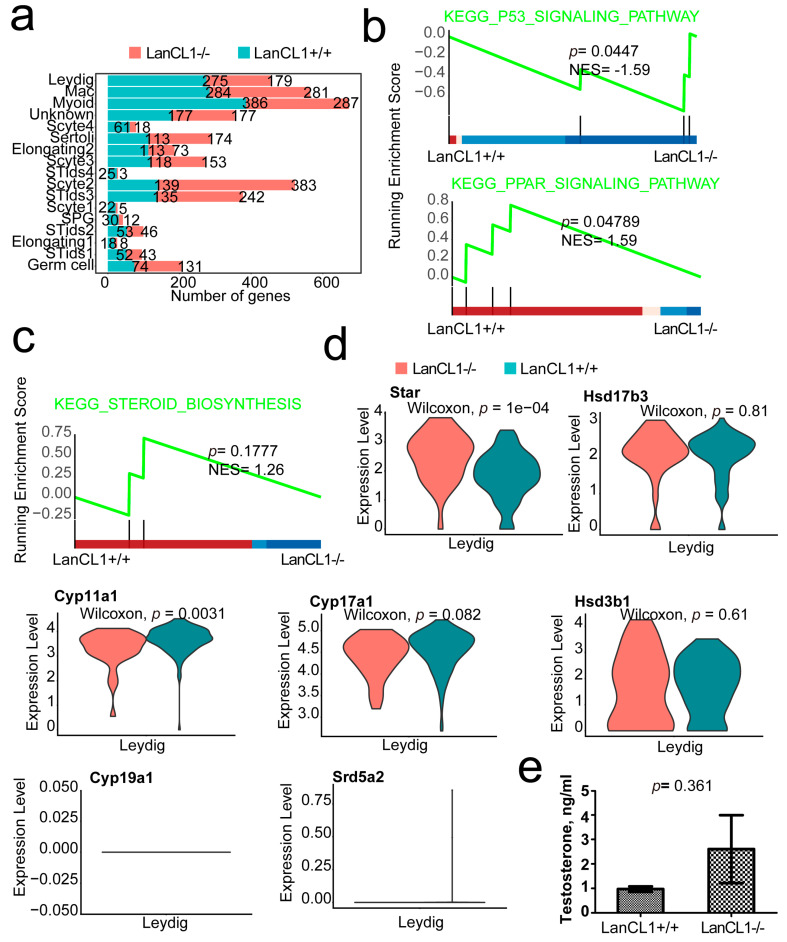
*LanCL1* gene regulated the *P53* and *PPAR* signaling other than the testosterone synthesis in the Leyding cells. (**a**) The numbers of DEGs in every cell clusters. (**b**) The GSEA of the DEGs in the *LanCL1*+/+ and *LanCL1−*/*−* mice of Leydig cell. (**c**) The steroid biosynthesis. (**d**) The marker genes of testosterone biosynthesis. (**e**) Serum testosterone level in the *LanCL1*+/+ and *LanCL1−*/*−* mice.

**Table 1 cells-11-04135-t001:** Results of fertility disturbed by the *LanCL1* gene.

Genotype of Mice		No. Male Mice	No. Female Mice	Pregnant Rate	Time of Vaginal Plug ^b^	*p* ^c^	Litter Size	*p* ^c^	No. Neonatal Male Mice	*p* ^c^	No. Neonatal Female Mice	*p* ^c^
Male	Female											
(+/+)	(+/+)	6	24	91.67% (22/24)	2.33 ± 0.16 ^a^	0.084	7.73 ± 0.41	0.027	3.81 ± 0.34	0.210	3.75 ± 0.52	0.467
(−/−)	(+/+)	6	24	91.67% (22/24)	3.00 ± 0.35		6.59 ± 0.28		3.30 ± 0.23		3.35 ± 0.25	

^a^ All the data was showed as mean ± SEM. ^b^ The deadline of the time of vaginal plug last to 10 days. ^c^ Some neonatal mice were dead. These cages were removed for the statistical analysis.

## Data Availability

The cellrager data were downloaded in https://www.jianguoyun.com/p/DaTbLREQ66iKCxjW1OIEIAA, accessed on 11 December 2022.

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
