# Peer review of "Single-Cell RNAseq Resolve the Potential Effects of LanCL1 Gene in the Mouse Testis"

_cells, 2022, doi:10.3390/cells11244135_

Round 1
Reviewer 1 Report
The manuscript appears well writren and the experiments well conducted. The focus of the study appears original.
I suggest to the authors only two modifications.
The role of LANCL1 gene and relative protein Is abountantly treated in the Discussione section together with the current literature in this regard. It could be more appropriate to treat this point in the Introduction section.
In the Discussione section, there Is a lack of the strenghts and weakness of the present study as well as how future experiments may amplify the knowledge in this field.
Author Response
Reviewer 1
Comments and Suggestions for Authors
The manuscript appears well writren and the experiments well conducted. The focus of the study appears original.
I suggest to the authors only two modifications.
Question 1: The role of LANCL1 gene and relative protein Is abountantly treated in the Discussione section together with the current literature in this regard. It could be more appropriate to treat this point in the Introduction section.
Answer: Thank you for your significant advice. We have moved some discussions of current literatures to the “Introduction” section. We hope that the new paper would be more fluent and rational.
Question 2: In the Discussione section, there Is a lack of the strenghts and weakness of the present study as well as how future experiments may amplify the knowledge in this field.
Answer: We have discussed the strengths and weakness of our study as well as how future experiments may amplify the knowledge in this field in the “Limitations” section.
Reviewer 2 Report
Cells
COMMENTS TO THE EDITORS AND THE AUTHORS
Manuscript ID cells-2047398: “Single-cell RNAseq reveal the protection of LANCL1 gene in the male infertility"
Dear Editors and Authors,
Please find below some of the comments for the above-mentioned manuscript.
SUMMARY OF THE CONTENT
The authors constructed transgenic LanCL1 knockout mice. The mating experiments, semen analysis and single-cell RNAseq of testicular tissue were performed. The authors stated that the mating experiment suggested that LanCL1 gene would significantly influence the reproduction ability of male mice. Results of single-cell RNAseq showed the high expression of LanCL1 gene in the spermatocytes and spermatids. The authors stated that the LanCL1 gene knockout could: promote M2 macrophage polarity, affect the germ cells and stromal cells, disturb the normal process of spermatogenesis. However, in the same samples the effects were absent in the steroidogenic genes.
THE OVERALL OPINION OF THE MANUSCRIPT
The strengths: the manuscript presents new and interesting knowledge; the results were obtained from the in vivo experiments; the figures very clearly present the results.
The limitations: the aim (“to discover the function of LanCL1 gene in the male reproduction”) was not achieved; the title, as well as the many sentences in the results section as well as the conclusions are not supported by data presented; the title, as well as the many parts of the text as well as the conclusions are not precisely formulated; the citation of the original, and important recent advances in the field focusing on the subject of the study are missing; the aim of the study is missing in the abstract; the intra- and inter- assay coefficients are not provided; the results of statistical analyses are in in collision with SEM.
Accordingly, the new experiments and significant changes of the text are required for further consideration. Please find enclosed some of the suggestions in the comments to the authors listed below.
(1) TITLE
Please consider modifying the title since the results are obtained on the mice and there are no results to prove the title. Namely, the results are obtained on null mice and there are no results to prove the title.
(2) ABSTRACT
2.1. Please state the aim of the study in the abstract.
2.2. Please modify the text of the abstract according to the results and figures obtained from the new experiments and measurements mentioned below.
(3) INTRODUCTION
3.1. Please describe the original and important recent advances in the field focusing on the subject of the study.
Just one of the examples is the citation of the “International Committee for Monitoring Assisted Reproductive Technology (ICMART) and the World Health Organization (WHO) revised glossary of ART terminology” from 2009.
3.2. The aim (“to discover the function of LanCL1 gene in the male reproduction”; page 2, line 48) was not achieved
3.3. Please precisely state the “reproductive experiments” (page 2, line 49).
3.4. The statement “Our study would elucidate the association between LanCL1 gene and male infertility on the single cell level“ (page 2, lines 50-51) is not correct since null mice were used.
(4) MATERIALS AND METHODS
4.1. Pleases preform mechanistic approach to show the direct effect of LanCL1 gene.
4.2. Please provide the Key resources table.
4.3. Please describe how do you calculated the number of animals to achieve accurate statistics.
4.4. Please provide photos of the gels showing the results of genotyping using the PCR analyses.
4.5. Please provide the levels of the hormones (at least androgens) to prove physiological significance.
4.6. Please provide the results of capacitation and acrosome reaction to prove physiological significance.
4.7. Please provide the amount of the protein used for western blot analyses.
4.8. Please provide the amount of RNA used for real-time PCR analysis.
4.9. Please provide the Ct values for the genes discussed.
4.10. Please provide intra- as well as inter-assay coefficients for all analyses.
(5) RESULTS
5.1. Please provide results and figures obtained from the new experiments and measurements mentioned above.
5.2. Please check the accuracy of the statistical analyses. Namely, there are huge visible discrepancies in relation to the SEM values.
5.3. Many parts of the text are not supported by the data. Some of the examples are listed below
Subtitle 3.2. LanCL1 gene directly influence the semen quality
Subtitle 3.3. LanCL1 gene suppress the expression of Cir1 gene in the spermatogenesis
and other subtitles.
(6) DISCUSSION
6.1. Please discuss the original, important and recent advances in the field focusing on the subject of the study.
6.2. Please discuss the results obtained from the new experiments and measurements mentioned above.
6.3. Please discuss the limitations of the study.
(7) CONCLUSIONS
7.1. Please use only results obtained in your study (the first and the last sentences are pointless).
7.2. Please modify the text of the conclusions according to the results and figures obtained from the new experiments and measurements mentioned above.
(8) REFERENCES
8.1. Please provide references describing the original, important and the recent advance in the field.
(9) FIGURES and FIGURE LEGENDS
9.1. Please remove Figure 1 since presents well know data obtained from the public library (https://www.proteinatlas.org/ENSG00000115365-LANCL1/tissue).
9.2. Please provide new figures and figure legends showing the new results obtained from the new experiments and measurements mentioned above.
I would greatly appreciate if you will contact me if you find something in my comments is missing/unclear/incorrect.
Good luck and all the best J
Author Response
(1) TITLE
Question 1: Please consider modifying the title since the results are obtained on the mice and there are no results to prove the title. Namely, the results are obtained on null mice and there are no results to prove the title.
Answer: Thank you for your suggestion. The title was revised as “Single-cell RNAseq resolve the potential effects of LanCL1 gene in the mouse testis”.
(2) ABSTRACT
Question 2: 2.1. Please state the aim of the study in the abstract.
Answer: Thank you for your advices. We have added the aim of the study in the “Abstract” section.
Question 3: 2.2. Please modify the text of the abstract according to the results and figures obtained from the new experiments and measurements mentioned below.
Answer: We have revised our abstract according to the results and figures obtained from the new experiments and measurements mentioned below.
(3) INTRODUCTION
Question 4: 3.1. Please describe the original and important recent advances in the field focusing on the subject of the study.
Just one of the examples is the citation of the “International Committee for Monitoring Assisted Reproductive Technology (ICMART) and the World Health Organization (WHO) revised glossary of ART terminology” from 2009.
Answer: This time, we have updated the data of our introduction with the recent studies.
Question 5: 3.2. The aim (“to discover the function of LanCL1 gene in the male reproduction”; page 2, line 48) was not achieved
Answer: According to our results, we have revised our aim as “In order to understand the potential functions of LanCL1 gene in the testis, this study was conducted in the transgenic LanCL1-/- mice with the single-cell RNA-seq, mouse breeding experiment and semen quality assessment applied.” in the “Introduction” section.
Question 6: 3.3. Please precisely state the “reproductive experiments” (page 2, line 49).
Answer: We are sorry for our confused description. The “reproductive experiments” mainly referred to the mouse breeding experiment. In the revised paper, we have defined the experiment clearly.
Question 7: 3.4. The statement “Our study would elucidate the association between LanCL1 gene and male infertility on the single cell level“ (page 2, lines 50-51) is not correct since null mice were used.
Answer: We are sorry for the improper sentence. We have revised it as “Our study would elucidate the potential effects of LanCL1 gene in various testicular cells, which might pave the way for the further mechanistic investigation of LanCL1 gene in the male infertility on the single-cell level.” in the “Introduction” section.
(4) MATERIALS AND METHODS
Question 8: 4.1. Pleases preform mechanistic approach to show the direct effect of LanCL1 gene.
Answer: In fact, Huang et al [1] also conducted a study about the function of LanCL1 gene in the male subfertility recently. In their study, they also found that LanCL1 gene deficiency was associated with the poor male fertility. After LanCL1 gene knockout, the mitochondria were swollen with reduced mitochondrial electron density, which resulted in the sperm motility defects. Considering the antioxidant activity of LanCL1 gene [2], they also found that LanCL1 gene could regulate the spermatogenic redox homeostasis to protect spermatogenesis from the oxidative damage. Once LanCL1 gene knockout, redox imbalance was observed in the testis with increased dihydroethidium-labeled ROS accumulation and decreased NADPH/NADP ratio. Moreover, increased LanCL1 gene expression could attenuate the defects in spermatozoal concentration, motility and forward motility induced by oxidative damage. According to these results, it was sure that there might be direct effect of LanCL1 gene in the male spermatogenesis through maintaining the redox homeostasis. However, no comprehensive landscape of all the testicular cell and cell Interactions after LanCL1 gene knockout on the single-cell level had been described. So, our study was conducted. Our results discovered some vital signal pathways in various single-cells to influence the spermatogenesis, which would pave a way for the further mechanism research on the single-cell level. In order to emphasize the features of our study, we had discussed their study in the “Introduction” and “Discussion” section. We also explained the limitation without more mechanistic results in our study in the “Limitations” paragraph.
- Huang, C.; Yang, C.; Pang, D.; Li, C.; Gong, H.; Cao, X.; He, X.; Chen, X.; Mu, B.; Cui, Y.; Liu, W.; Luo, Q.; Cheng, A.; Jia, L.; Chen, M.; Xiao, B.; Chen, Z. Animal models of male subfertility targeted on LanCL1-regulated spermatogenic redox homeostasis. Lab. Anim (NY). 2022,51,133-145.
- Huang, C.; Chen, M.; Pang, D.; Bi, D.; Zou, Y.; Xia, X.; Yang, W.; Luo, L.; Deng, R.; Tan, H.; Zhou, L.; Yu, S.; Guo, L.; Du, X.; Cui, Y.; Hu, J.; Mao, Q.; Worley, P.F.; Xiao, B. Developmental and activity-dependent expression of LanCL1 confers antioxidant activity required for neuronal survival. Dev. Cell. 2014,30,479-487.
Question 9: 4.2. Please provide the Key resources table.
Answer: We have added the Key resources table in the supplementary materials.
Question 10: 4.3. Please describe how do you calculated the number of animals to achieve accurate statistics.
Answer: Thank you for your significant question. In this study, in order to acquire relatively stable statistical differences, at least 3 paired mice were used in the experiments, such as qPCR. As for the mouse breeding experiment, we tried to achieve more accurate statistics with much more mice. So, in this study, 24 paired female mice were used. Then, PASS software (V15.0) was used to calculate the minimum number of animals to achieve accurate statistics. In the calculation, the parameters were set as: Power=0.90, Alpha=0.05, Group allocation= Equal (N1=N2), μ1=6, μ2=4, ?=1.00. Finally, 7 paired samples were needed on this condition. So, the number of mice should be enough to achieve the right result.
Question 11: 4.4. Please provide photos of the gels showing the results of genotyping using the PCR analyses.
Answer: We have added the photos of the gels showing the results of genotyping using the PCR analyses in the Supplementary Figure 1. Meanwhile, we also described the identification method of genotype in the “Methods and Materials” section.
Question 12: 4.5. Please provide the levels of the hormones (at least androgens) to prove physiological significance.
Answer: Thank you for your suggestion. We have compared the level of androgens in both the homozygous LanCL1 gene knockout mice (LanCL1-/-) and wide type (LanCL1+/+) male mice with the enzyme-linked immunosorbent assay (ELISA). The result also confirmed no significant influences of LanCL1 gene in the level of testosterone. The figure was shown in the Figure 5e. The method of detection and result were added in the “Methods and Materials” and “Results” section.
Question 13: 4.6. Please provide the results of capacitation and acrosome reaction to prove physiological significance.
Answer: In the previous study [1], spermatids’ acrosome was absent with the LanCL1 gene expression, which suggested that LanCL1 gene might not mainly influence the capacitation and acrosome reaction of spermatids, but the mitochondrial damage, oxidative stress and germ cell apoptosis. In the revised paper, we had discussed in the “Discussion” section.
- Huang, C.; Yang, C.; Pang, D.; Li, C.; Gong, H.; Cao, X.; He, X.; Chen, X.; Mu, B.; Cui, Y.; Liu, W.; Luo, Q.; Cheng, A.; Jia, L.; Chen, M.; Xiao, B.; Chen, Z. Animal models of male subfertility targeted on LanCL1-regulated spermatogenic redox homeostasis. Lab. Anim (NY). 2022,51,133-145.
Question 14: 4.7. Please provide the amount of the protein used for western blot analyses.
Answer: We have added the Mean±SEM for amount of the protein used for western blot analyses of LanCL1 gene in the “Results” section.
Question 15: 4.8. Please provide the amount of RNA used for real-time PCR analysis.
Answer: We have added the Mean±SEM for the real-time PCR analysis of LanCL1 gene in the “Results” section.
Question 16: 4.9. Please provide the Ct values for the genes discussed.
Answer: The Ct values of the LanCL1 gene were presented in the following table.
Sample Name |
Gene Name |
Cq |
Cq Mean |
Cq Error |
2(-â–³Ct) |
mean for 2(-â–³Ct) of KOs |
2(-△△Ct) |
KO-/- 1 |
Lancl1 |
22.04 |
21.88667 |
0.138684 |
0.017701 |
0.020530036 |
0.862215 |
KO-/- 1 |
Lancl1 |
21.85 |
21.88667 |
0.138684 |
0.020193 |
0.983584 |
|
KO-/- 1 |
Lancl1 |
21.77 |
21.88667 |
0.138684 |
0.021344 |
1.039666 |
|
KO-/- 1 |
Gapdh |
16.22 |
16.22 |
0.01 |
|||
KO-/- 1 |
Gapdh |
16.21 |
16.22 |
0.01 |
|||
KO-/- 1 |
Gapdh |
16.23 |
16.22 |
0.01 |
|||
KO-/- 2 |
Lancl1 |
19.44 |
19.44333 |
0.005774 |
0.019505 |
0.950079 |
|
KO-/- 2 |
Lancl1 |
19.44 |
19.44333 |
0.005774 |
0.019505 |
0.950079 |
|
KO-/- 2 |
Lancl1 |
19.45 |
19.44333 |
0.005774 |
0.01937 |
0.943517 |
|
KO-/- 2 |
Gapdh |
13.77 |
13.76 |
0.017321 |
|||
KO-/- 2 |
Gapdh |
13.74 |
13.76 |
0.017321 |
|||
KO-/- 2 |
Gapdh |
13.77 |
13.76 |
0.017321 |
|||
KO-/- 3 |
Lancl1 |
19.82 |
19.74667 |
0.190875 |
0.021148 |
1.030102 |
|
KO-/- 3 |
Lancl1 |
19.53 |
19.74667 |
0.190875 |
0.025856 |
1.259444 |
|
KO-/- 3 |
Lancl1 |
19.89 |
19.74667 |
0.190875 |
0.020146 |
0.981314 |
|
KO-/- 3 |
Gapdh |
14.17 |
14.25667 |
0.109697 |
|||
KO-/- 3 |
Gapdh |
14.38 |
14.25667 |
0.109697 |
|||
KO-/- 3 |
Gapdh |
14.22 |
14.25667 |
0.109697 |
|||
WT1 |
Lancl1 |
19.42 |
19.56667 |
0.262742 |
0.106333 |
5.179406 |
|
WT1 |
Lancl1 |
19.41 |
19.56667 |
0.262742 |
0.107073 |
5.215432 |
|
WT1 |
Lancl1 |
19.87 |
19.56667 |
0.262742 |
0.077841 |
3.791547 |
|
WT1 |
Gapdh |
16.17 |
16.18667 |
0.066583 |
|||
WT1 |
Gapdh |
16.13 |
16.18667 |
0.066583 |
|||
WT1 |
Gapdh |
16.26 |
16.18667 |
0.066583 |
|||
WT2 |
Lancl1 |
20.44 |
20.45 |
0.01 |
0.045965 |
2.238895 |
|
WT2 |
Lancl1 |
20.46 |
20.45 |
0.01 |
0.045332 |
2.208071 |
|
WT2 |
Lancl1 |
20.45 |
20.45 |
0.01 |
0.045647 |
2.22343 |
|
WT2 |
Gapdh |
16.01 |
15.99667 |
0.015275 |
|||
WT2 |
Gapdh |
15.98 |
15.99667 |
0.015275 |
|||
WT2 |
Gapdh |
16 |
15.99667 |
0.015275 |
|||
WT3 |
Lancl1 |
21.49 |
21.49 |
0.03 |
0.05329 |
2.595693 |
|
WT3 |
Lancl1 |
21.46 |
21.49 |
0.03 |
0.054409 |
2.650234 |
|
WT3 |
Lancl1 |
21.52 |
21.49 |
0.03 |
0.052193 |
2.542275 |
|
WT3 |
Gapdh |
17.28 |
17.26 |
0.017321 |
|||
WT3 |
Gapdh |
17.25 |
17.26 |
0.017321 |
|||
WT3 |
Gapdh |
17.25 |
17.26 |
0.017321 |
|||
KO+/- 1 |
Lancl1 |
21.72 |
21.75333 |
0.028868 |
0.023574 |
1.148263 |
|
KO+/- 1 |
Lancl1 |
21.77 |
21.75333 |
0.028868 |
0.022771 |
1.109149 |
|
KO+/- 1 |
Lancl1 |
21.77 |
21.75333 |
0.028868 |
0.022771 |
1.109149 |
|
KO+/- 1 |
Gapdh |
16.35 |
16.31333 |
0.032146 |
|||
KO+/- 1 |
Gapdh |
16.29 |
16.31333 |
0.032146 |
|||
KO+/- 1 |
Gapdh |
16.3 |
16.31333 |
0.032146 |
|||
KO+/- 2 |
Lancl1 |
21.2 |
21.25 |
0.05 |
0.023142 |
1.127234 |
|
KO+/- 2 |
Lancl1 |
21.3 |
21.25 |
0.05 |
0.021592 |
1.051746 |
|
KO+/- 2 |
Lancl1 |
21.25 |
21.25 |
0.05 |
0.022354 |
1.088836 |
|
KO+/- 2 |
Gapdh |
15.75 |
15.76667 |
0.020817 |
|||
KO+/- 2 |
Gapdh |
15.76 |
15.76667 |
0.020817 |
|||
KO+/- 2 |
Gapdh |
15.79 |
15.76667 |
0.020817 |
|||
KO+/- 3 |
Lancl1 |
21.65 |
21.60333 |
0.022561 |
0.023311 |
1.098946 |
|
KO+/- 3 |
Lancl1 |
21.6 |
21.60333 |
0.023357 |
1.1377 |
||
KO+/- 3 |
Lancl1 |
21.56 |
21.60333 |
0.024014 |
1.169685 |
||
KO+/- 3 |
Gapdh |
16.16 |
16.18 |
||||
KO+/- 3 |
Gapdh |
16.2 |
16.18 |
||||
KO+/- 3 |
Gapdh |
16.18 |
16.18 |
Question 17: 4.10. Please provide intra- as well as inter-assay coefficients for all analyses.
Answer: We have added the intra- as well as inter-assay coefficients for all the analyses.
(5) RESULTS
Question 18: 5.1. Please provide results and figures obtained from the new experiments and measurements mentioned above.
Answer: We have added the new results and figures in the “Results” section.
Question 19: 5.2. Please check the accuracy of the statistical analyses. Namely, there are huge visible discrepancies in relation to the SEM values.
Answer: Thank you for your significant reminders. We have checked all the statistical analyses again. And we found that the data were displayed with the Mean±SD instead of SEM. This time, we had revised them to be Mean±SEM. However, it is true that some data in our analyses seemed to have high variable coefficient (CV). These might be resulted from our experimental operations. Even though, combining with our all analyses, the results also could hinted the important role of LanCL1 gene in the function of testicular tissue related to the fertility, which would pave a way for the further mechanism research on various cell types. In order to emphasize the defects of our analyses, we discussed it in the “Limitations” section.
Question 20: 5.3. Many parts of the text are not supported by the data. Some of the examples are listed below
Subtitle 3.2. LanCL1 gene directly influence the semen quality
Subtitle 3.3. LanCL1 gene suppress the expression of Cir1 gene in the spermatogenesis
and other subtitles.
Answer: We are sorry for the improper subtitles in the “Results” section. We have revised the subtitles according to our data.
(6) DISCUSSION
Question 21: 6.1. Please discuss the original, important and recent advances in the field focusing on the subject of the study.
Answer: As an important study about the association between LanCL1 gene and subfertility, Huang et al had discussed the potential mechanisms about redox homeostasis of LanCL1 gene in the transgenic mice. Many results were consistent with ours. So, we have discussed their study in the “Discussion” section, for example, “Recently, Huang et al [1] performed a study about the LanCL1-regulated spermatogenic redox homeostasis, which had described the mechanisms of LanCL1 gene in influencing the male subfertility. Once LanCL1 gene knockout, redox imbalance was observed in the testis with increased dihydroethidium-labeled ROS accumulation and decreased NADPH/NADP ratio. In their study, they found the same results with ours about the decreased sperm motility and poor male fertility in the LanCL1 gene knockout mice. The attenuated motility and fertility might result from the mitochondrial damage, oxidative stress and germ cell apoptosis, other than the acrosome reaction of spermatids for the absent LanCL1 gene expression. Moreover, conditional increased LanCL1 gene expression could attenuate the defects in spermatozoal concentration, motility and forward motility induced by oxidative damage.”, “In the LanCL1 knockout mice, the ROS and inflammation increased [1]. In order to maintain body homeostasis, the anti-inflammatory macrophage would increase. Consistent with our results, the gene markers of M2-type macrophage was also highly expressed in the LanCL1-/- mice.” and “Although as an important cell in the testosterone synthesis, most of the key rate-limiting enzymes of testosterone synthesis such as Cyp17a1, Cyp19a1, Hsd3b1, Srd5a1, Srd5a2 and Hsd17b3, were not presented remarkable difference in the LanCL1 knockout mice. Moreover, the serum testosterone level had also no statistic differences, which was also consistent with the results of Huang et al [1.So, LanCL1 gene would not influence the function of testosterone synthesis in Leydig cell. ”.
- Huang, C.; Yang, C.; Pang, D.; Li, C.; Gong, H.; Cao, X.; He, X.; Chen, X.; Mu, B.; Cui, Y.; Liu, W.; Luo, Q.; Cheng, A.; Jia, L.; Chen, M.; Xiao, B.; Chen, Z. Animal models of male subfertility targeted on LanCL1-regulated spermatogenic redox homeostasis. Lab. Anim (NY). 2022,51,133-145.
Question 22: 6.2. Please discuss the results obtained from the new experiments and measurements mentioned above.
Answer: In the previous study, spermatids’ acrosome was absent with the LanCL1 gene expression, which suggested that LanCL1 gene might not mainly influence the capacitation and acrosome reaction of spermatids, but the mitochondrial damage, oxidative stress and germ cell apoptosis. In the revised paper, we had discussed in the “Discussion” section. Additionally, testosterone synthesis would not be influenced by LanCL1 gene, which was also consistent with the previous study [1]. In the revised paper, we had discussed in the “Discussion” section.
- Huang, C.; Yang, C.; Pang, D.; Li, C.; Gong, H.; Cao, X.; He, X.; Chen, X.; Mu, B.; Cui, Y.; Liu, W.; Luo, Q.; Cheng, A.; Jia, L.; Chen, M.; Xiao, B.; Chen, Z. Animal models of male subfertility targeted on LanCL1-regulated spermatogenic redox homeostasis. Lab. Anim (NY). 2022,51,133-145.
Question 23: 6.3. Please discuss the limitations of the study.
Answer: We have discussed the limitations of the study in the “Limitations” section. However, some limitations should also be emphasized as follows: 1) some results of experiments such as semen analysis and testosterone level seemed to be variable in different mice. So, repetitive experiments with much more mice should be considered in the further studies; 2) results of detailed mechanism were scarce in our study to verify the discovery of single-cell RNAseq.
(7) CONCLUSIONS
Question 24: 7.1. Please use only results obtained in your study (the first and the last sentences are pointless).
Answer: Thank you for your suggestions. We have rewritten our “Conclusion” section according to our own results.
Question 25: 7.2. Please modify the text of the conclusions according to the results and figures obtained from the new experiments and measurements mentioned above.
Answer: Thank you for your suggestions. We have rewritten our “Conclusion” section and added the new results of serum testosterone level.
(8) REFERENCES
Question 26: 8.1. Please provide references describing the original, important and the recent advance in the field.
Answer: We have updated the references describing the original, important and the recent advance in the field.
(9) FIGURES and FIGURE LEGENDS
Question 27: 9.1. Please remove Figure 1 since presents well know data obtained from the public library (https://www.proteinatlas.org/ENSG00000115365-LANCL1/tissue).
Answer: We have removed the figure in the revised paper.
Question 28: 9.2. Please provide new figures and figure legends showing the new results obtained from the new experiments and measurements mentioned above.
Answer: Thank you for your significant advices. We have added the new figure and figure legends in the revised paper.
Round 2
Reviewer 2 Report
Dear the Editors and the Authors,The manuscript has been sufficiently improved to warrant publication in Cells.
All the best!